# A framework for identifying calcium accumulation problem in cropland: Integrating field surveys, legacy soil map, and machine learning models

Xingjie Yin[1☉], Haile Zhao[1☉], Yuchao Luo[1☉], Yuling Jin[1☉], Zhihua Pan[2☉], Wenting Liu[1☉], Pingli An[1]*

1 College of Land Science and Technology, China Agricultural University, Beijing, China, 2 College of Resources and Environmental Science, China Agricultural University, Beijing, China

Present address: College of Land Science and Technology, China Agricultural University, No. 2 Yuanmingyuan West Road, Haidian, Beijing 100193, China.
☉ These authors contributed equally to this work.
* anpl@cau.edu.cn

## Abstract

The calcium accumulation problem (CAP) in cinnamon soil regions of northern China significantly impacts crop yields. Identifying and mitigating CAP is crucial for improving soil quality and agricultural productivity. This study, based on field research in Aohan Banner, Chifeng City, utilizes legacy soil maps to construct a CAP dataset and evaluates the predictive performance of several machine learning models. The influence of topography on CAP is also analyzed. Key findings include: (1) In the study area, CAP predominantly manifests as block formations in dry land. Of the surveyed farmers, 58% report CAP in their cropland, with 84% noting reduced yields, though 76% have not implemented any specific mitigation measures. (2) Evaluation of machine learning models shows that tree-based models (BRT and XGBoost) outperform others in predicting CAP, with BRT demonstrating superior mapping capabilities. (3) Spatial analysis reveals that CAP is more common in the eastern and central regions of Aohan Banner, particularly in terrains such as slopes, ridges, and peaks. Additionally, the cold-to-hot zone ratio increases significantly as terrain transitions from dry to humid. (4) Regression analysis shows a strong negative correlation between terrain variables (e.g., MRVBF and GEO) and the likelihood of CAP. A further analysis indicates that CAP is more likely to occur in areas with higher soil erosion risk. These findings provide valuable insights for identifying CAP in regional soil mapping and for guiding future research in this area.

## Introduction

Cinnamon soils, a globally distributed soil type, are mainly found in Liaohe Basin of China [1]. They typically form on lime-rich loess parent material and are shaped by

**Data availability statement:** Data cannot be shared publicly because of a confidentiality agreement was signed when applying for soil data. Data are available from China Soil Science Data Center Institutional Data Access/Ethics Committee (contact via 025-86881307) for researchers who meet the criteria for access to confidential data. The data underlying the results presented in the study are available from National earth system science data center, soildatacenter@issas.ac.cn.

**Funding:** This work was supported by the National Key Research and Development Plan of China (2022YFD1500602-3) and the National Natural Science Foundation of China (grant number 42271268). The funders had no role in study design, data collection and analysis, decision to publish, or preparation of the manuscript.

**Competing interests:** The authors have declared that no competing interests exist.

a semi-humid to semi-arid monsoon climate [2]. This environment drives key soil-forming processes, including the leaching and accumulation of calcium carbonate, which lead to the development of significant amounts of pedogenic carbonates (PC) within the soil profile.

PC is deposited in various forms (including earthworm biospheroliths, rhizoliths, pseudomycels, nodules, coatings and calcrete) through the dissolution, movement and re-precipitation of calcium carbonate [3]. Its distribution is influenced by multiple factors, including climate [4–6], topography [7–10], human activities [11–14], etc. A sediment layer of a certain depth is formed through eluviation and illuviation of PC, which varies with precipitation [11,15]. Land use changes, especially the conversion of natural landscapes to cropland, often lead to increased PC content [11,13,16], with a tendency for accumulation near the surface [17,18]. Irrigation can eluviate PC to deeper layers [17,19], while in arid regions without additional water inputs, PC tends to accumulate in the shallow layers of croplands [20,21].

Calcium accumulation problem (CAP) caused by excessive PC accumulation, which affects the physical, chemical and biological properties of the soil [22], and thus indirectly affects crop productivity. Such as clogging soil pores [23], weakening inter-root water infiltration and movement [24], reducing plant utilization of micronutrients [15], and threatening sustainable agricultural development. Thus, identifying and managing CAP is central to cinnamon soil cropland remediation research.

Legacy soil map (LSM) is acknowledged to contain important information on the spatial distribution of soil classes [25,26]. LSM also contain detailed profile descriptions that reflect the conditions of the soil at the time of mapping. Thus, LSM-based soil characteristic prediction has become a key research focus [25–29]. While various machine learning models are available for predicting soil characteristics, there are fewer studies that have focused on modeling data derived from LSM [30–35]. Unlike real soil samples, LSM data lacks stability, which presents challenges for accurate modeling. Therefore, it is crucial to identify which models can reliably predict CAP using LSM data.

In this study, we aim to address the following objectives:

(1) Construct a CAP characteristic dataset through field research and LSM sampling;

(2) Evaluate the performance of different machine learning models in identifying the spatial distribution of CAP in cinnamon soil croplands;

(3) Analyze the factors influencing CAP formation.

The results of this paper have important implications for improving the quality of cinnamon soil croplands and informing policy development.

## Materials and methods

### Study area

There are 47,000 km² cropland in cinnamon soil area from Liaohe Plain, where there are widespread problems such as drought, low rainfall, serious soil erosion, and obvious calcium deposits. Aohan banner is a typical area in Liaohe Plain, located in the southeastern quadrant of Chifeng City, Inner Mongolia (Fig 1b). Aohan Banner's

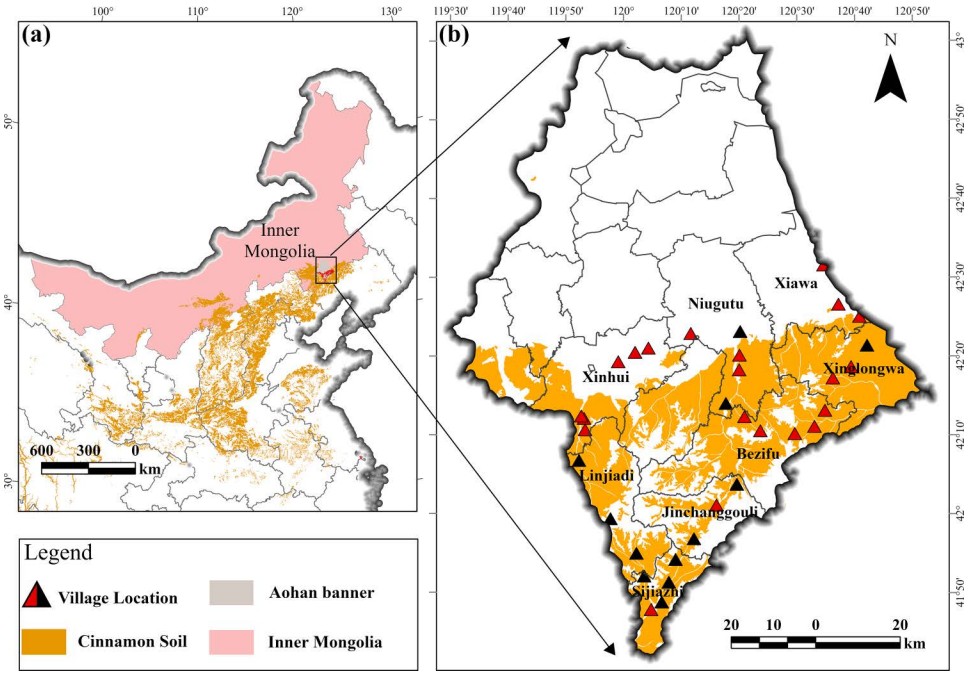

**Fig 1. Location Map of Aohan Banner.** All map boundary data in this paper are the same and are freely available at the National Platform for Common GeoSpatial Information Services 'www.tianditu.gov.cn'. The review map number is GS (2024) 0650, and the latter map is not repeated.

landscape features a gentle and uneven slope, descending from the southeast to the northwest. This region experiences a mid-temperate semi-arid continental monsoon climate, characterized by frequent droughts, annual rainfall varying between 310–460 mm, and marked temperature variations between day and night. With an average annual temperature of 6.9°C and a frost-free period lasting 138 days, the area supports the cultivation of various crops including corn, grain, sorghum, potatoes, and soybeans.

The area is administratively divided into 17 townships. This study focuses on eight townships in the southern region of Aohan. The townships are as follows: Niugutu, Xiawa, Xinhui, Xinglongwa, Bezifu, Jinchanggouliang, Linjiadi, and Sijiazhi. As of 2022, Aohan Banner covers a total area of 8,300 square kilometers, is home to a population of 597,600, generates a GDP of 17.73 billion yuan, and maintains 4 million acres of cropland, contributing to a grain output of 2.55 billion pounds.

## Principles of identifying CAP

CAP is defined as a soil characteristic that negatively impacts crops by degrading the physical and chemical properties of the soil due to the accumulation of carbonates. PC is the primary component of these accumulated carbonates, with calcium carbonate being the main constituent of PC [3]. The essence of CAP lies in the accumulation of PC, therefore factors influencing PC formation are key contributors to CAP, like parent material, topography, climate and human activity, etc.

The different spatial distribution of factors plays a crucial role in identifying CAP. The different types of cinnamon soil have similar parent material, providing a rich source of calcium carbonate, but with low spatial variability; Aohan Banner is an arid and semi-arid region with an average annual rainfall of less than 360 mm, which makes the area "suitable" for the formation of CAP in terms of rainfall; However, the spatial distribution of mean annual rainfall varies slightly across the Aohan Banner at the county scale. Therefore, the soil profile is rich in calcium carbonate from the parent material, which lays a "good" material foundation for the formation of CAP, where climate factors create an environmental condition. However, both of them not the primary cause of the different distribution patterns of calcium deposits in the region.

Differences in topography affect the clustering and dispersal of surface materials, playing a dominant role in the differential distribution of CAP within the county. The formation of the CAP is largely due to insufficient water infiltration, which prevents calcium carbonate in the soil from leaching to deeper layers. Therefore, topographic variables are used as environmental covariates to in this study to predict CAP.

### LSM sampling

The county-level LSM scale is 1:50,000, providing an accurate spatial distribution of various soil types for sampling. Before LSM sampling, we validated the soil map using data from 33 field-dug soil profiles. (Fig 2) illustrates some of the profiles, with the phenomenon of CAP, forming points, blocks, and surfaces. For each profile, we recorded the actual observed soil type, coordinates, and CAP labeling information (S1 Table). The LSM was compared to the actual soil types, yielding an overall accuracy of 94%, confirming the reliability of the LSM.

Typical profile descriptions for each subgroup on the map were derived from the Soil Census (Table 1). When "carbonate deposits" were observed in a typical soil profile, it is assumed that CAP was present in the corresponding soil type. (Fig 3a) illustrates the distribution of soil types in southern Aohan Banner, where 628 of 1106 cinnamon soil polygons exhibit CAP. The number of sampling points per cinnamon soil polygon was determined based on area as weight and randomly sampled. A total of 10,000 sample points were allocated, some of which are shown in (Fig 3b).

### Correlation of cinnamon soil with the World Reference Base (WRB)

Cinnamon soil is a zonal soil found in the arid forest and shrub-steppe zone of the warm temperate region. To ensure the results of this study applicable to similar soils globally, the correlation between cinnamon soils and WRB classification was examined (Table 2). The results indicate that cinnamon soils have the highest referability to Cambisols (48%) and 61% for Castanozeras with Kastanozems.

Cinnamon soil in Aohan banner is divided into five subclasses: coarse-grained cinnamon soil, leached cinnamon soil, typical cinnamon soil, calcareous cinnamon soil, and hydromorphic cinnamon soil. Calcareous cinnamon soil occupies the largest area, comprising 61.02% of the total cinnamon soil area in Aohan Banner. In the central part of Aohan Banner, the soil type transitions from calcareous cinnamon soil to Castanozeras. Local calcareous cinnamon soils are difficult to distinguish from Castanozeras due to the similarities in both properties and the influence of soil erosion. The subclasses of cinnamon soils that contain CAP are limited to calcareous cinnamon soil (92.3% of the area) and typical cinnamon soil

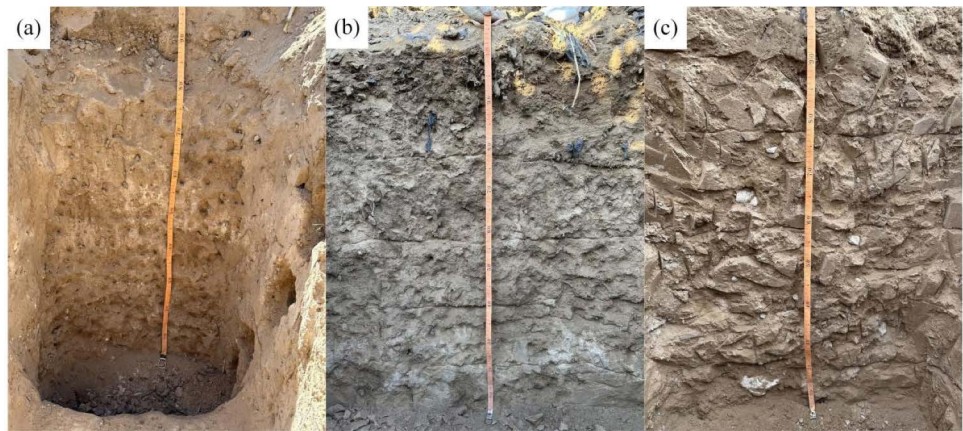

**Fig 2. A part of soil profiles.**

(7.7% of the area). Thus, it is possible to approximate the Kastanozems in the WRB classification with reference to the cinnamon soils described in this study.

### Machine learning model

**Covariates.** In summary, spatial dissimilarity of variables is key to identification. Thus, we only wish to consider the soil physical and chemical properties, such as calcium carbonate content, soil pore size, etc. These properties manifest themselves differently in different terrains. We therefore identified CAP by considering topographic factors to indirectly consider the combined expression of soil properties.

Topographic variables with significant spatial differentiation are selected as covariates in this paper (Table 3). These covariates characterize the relief and morphology of the terrain (DEM, Planc, FC, VD, Con), hydrologic terrain features (TWI, MRVBF), the effect of terrain relief on water flow (Planc, FC, Con), and geomorphic patterns (GEO). All topographic variables were calculated in SAGA, with a resolution of 12.5m.

**Table 1. Cinnamon soil typical profile description.**

| Great groups | Subgroups | Parent materials | Typical profile description |
|---|---|---|---|
| Cinnamon loess soil | Non-eroded in flat terrain | Loess or loess-like material | At 41 cm: Lime reaction and pseudomycels deposition. |
| | Lightly eroded | | At 38 cm: Lime reaction; At 60 cm and deeper: pseudomycels deposition |
| | Moderately eroded | | At 37 cm: Lime reaction; pseudomycels deposition in the lower profile. |
| | Severely eroded | | At 38 cm: Lime reaction; At 42 cm: Calcium carbonate deposition. |
| | Extremely eroded | | At 37 cm: Lime reaction and pseudomycels deposition |
| Calcareous cinnamon soil | Slightly gravelly medium-textured | Limestone | At 50 cm: Calcium carbonate deposition. |
| | Gravelly thin-textured | | Below horizon A: Large amounts of calcium carbonate deposition. |
| | Slightly gravelly medium-textured | | At 30 cm: White-dry layer. |
| Cinnamon loamy sand soil | Slightly gravelly thick-textured | Loess or loess-like material | At 39.5 cm below surface: Calcium carbonate deposition |
| Calcareous cinnamon red soil | Lightly eroded | Red Soil | At 36 cm: Calcium carbonate deposition. |
| | Moderately eroded | | At 19 cm: Calcium carbonate deposition. |
| | Severely eroded | | At 40 cm: White-dry layer; lime nodules on the surface. |
| Calcareous cinnamon red loess soil | Lightly eroded | Red loess soil | Throughout the profile: Lime reaction; pseudomycels deposition. |
| | Moderately eroded | | At 35 cm: Calcium carbonate deposition begins. |
| | Severely eroded | | Between loess and red soil: White-dry lime layer. |
| Calcareous cinnamon loess soil | Non-eroded in flat terrain | Loess or loess-like material | At 39 cm: Calcium carbonate deposition. |
| | Lightly eroded | | At 33 cm: Calcium carbonate deposition begins; pseudomycels deposition starts at 40 cm. |
| | Moderately eroded | | At 33 cm: Calcium carbonate deposition. |
| | Severely eroded | | At 29 cm: Calcium carbonate deposition; lime reaction throughout the profile. |
| | Extremely eroded | | At 30 cm: Calcium carbonate deposition. |

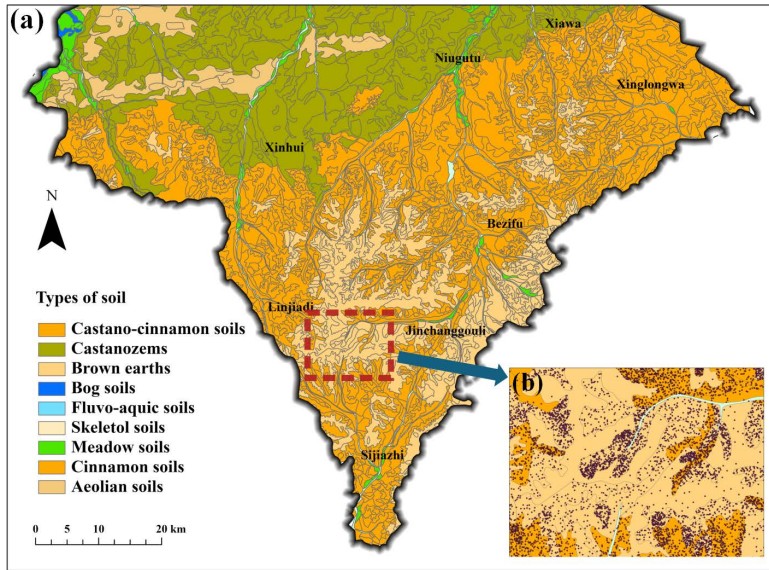

**Fig 3. Legacy soil map in Aohan banner. (a)**Distribution of different soils in Aohan Banner. **(b)**Example of LSM sampling.

**Table 2. Relevance between cinnamon soil and the WRB Group.**

| Soil order | Soil great group | Main forming process and morphological feature | WRB Group | MR (%) |
|---|---|---|---|---|
| Semi-alfisols | Cinnamon soils | Semi-hunid warm temperate zone, argillification and calcium sediment caused by leaching, with saturated base, dark brown B horizon | Cambisols | 48 |
| Pedo-cals | Castano-zeras | Temperate semi-arid grassland, having nut-brown humus horizon and grey-white calcic horizon | Kasta-nozems | 61 |

MR, Maximum referability of WRB soil Group to China Soil Great Group [36]

**Model selection.** For the purpose of resolving linear or non-linear relationships between terrain variables, we chose three common models (XGBoost, BRT and SVM, with decreasing model complexity in that order) to account for non-linear relationships between terrain variables, and Logistic regression to account for linear relationships between terrain variables (Table 4).

**XGBoost** is an efficient integration method for dealing with complex nonlinear relationships and large-scale data, with high computational efficiency and the ability to prevent overfitting [45]. **Boosted regression trees** (BRT) is another integrated learning method that improves the performance of regression tasks by combining multiple decision tree models. BRT can effectively deal with nonlinear relationships and interactions between variables [46]. **SVM**, on the other hand, handles classification and regression tasks by finding the optimal hyperplane, which is especially suitable for high-dimensional data and nonlinear relationships [47]. **Logistic regression** makes binary predictions by converting linear regression outputs to probability values using a sigmoid function. These models were constructed using the relevant R language packages (e.g., gbm, xgboost, svm, glm). Parameters were set according to specific task to ensure model accuracy and stability (Table 4).

**Accuracy verification methods.** To evaluate the accuracy of the model's predictions, we used three metrics: Area Under the Curve (AUC), Logarithmic Loss (LL), and Overall Accuracy (OA).

(1) AUC

**Table 3. Covariates Description.**

| Variables | Shortening | Description | Ref. |
|---|---|---|---|
| Digital Elevation Model | DEM | Elevation information for regional terrain | |
| Topographic Wetness Index | TWI | Topography-driven changes in soil moisture | [37–39] |
| Multi-resolution Valley Bottom Flatness | MRVBF | Identify areas that are flat and low relative to their surroundings | [40] |
| Geomorphons | GEO | Terrain classification based on machine vision principles | [41] |
| Plane curvature | Planc | The concave-convex pattern of slopes along the horizontal | [42] |
| Valley depth | VD | Difference between elevation and interpolated ridge level | |
| Flowline curvature | FC | The curvature of the water flow in the direction of the slope | [43] |
| Convexity | Con | Degree of convexity with respect to the mean horizontal plane | [44] |

**Table 4. Parameter selection for different models.**

| Model | Parameters |
|---|---|
| BRT | Family = bernoulli; learning rate = 0.005; bag fraction = 0.5; tree complexity = 10; |
| XGboost | max_depth = 10, eta = 0.01, nrounds = 500, objective = "binary:logistic", eval_metric = "auc" |
| SVM | kernel = "radial", cost = 1, scale = TRUE |
| LR | family = binomial |

The ROC curve is used to plot the true positive rate (TPR) and false positive rate (FPR) at different classification thresholds [48]. The area under the ROC curve, or AUC, represents the total area beneath the curve from (0,0) to (1,1). A higher AUC value indicates better classification accuracy, with an AUC greater than 0.7 considered indicative of a good model [49].

$$TPR = \frac{TP}{(TP + FN)}$$

$$FPR = \frac{FP}{(FP + TP)}$$

Where, TP denotes a true example, FP denotes a false positive example, and FN denotes a false negative example.

(2) LL

Logarithmic Loss is a loss function used to assess the accuracy of classification models. It measures the difference between the predicted probabilities and the actual category labels.

$$LL = -\frac{1}{N}[y_i \log(p_i) + (1 - y_i) \log(1 - p_i)]$$

Where, $N$ is the number of samples; $y_i$ is the true label of $i$, which in this paper is 0 or 1, representing the presence or absence of CAP; $p_i$ is the probability that the model predicts that the sample $i$ belongs to category 1; and $1 - p_i$ is the probability that the model predicts that the sample $i$ belongs to category 0.

(3) OA

OA is the proportion of samples correctly predicted by the model and is used to describe the proportion of total samples correctly classified by the model.

$$OA = \frac{TP + TN}{TP + TN + FP + FN}$$

**Precision-Recall curve.** In practical applications, it is essential to compare the model's predicted results with the true labels to assess its accuracy. However, the model's predictions typically yield continuous probabilities that need to be converted into discrete classifications based on a threshold. Therefore, identifying the optimal threshold is crucial for transforming these probabilities into final classification labels. This conversion allows us to calculate the model's accuracy and evaluate its performance on unseen data.

Selecting the threshold involves considering multiple factors. To maximize the identification of true positives, this study prioritizes high recall and precision. A high recall ensures that the model identifies as many true positives as possible—here, it refers to locations with CAP that are less likely to be missed. Consequently, the optimal threshold is determined using the Precision-Recall curve, which illustrates the trade-off between precision and recall at various thresholds.

### Data

(1) Survey data. The collection of data and information for this study was conducted in the first half of August 2023, with the survey period specifically spanning from August 6 to August 9, 2023. The questionnaire consisted of both closed and open-ended sections. The research involved 33 villages in Aohan Banner and 90 questionnaires were collected. The questionnaire consisted of three parts: 1) farmers' knowledge of CAP; 2) the impact of CAP in cropland on farmers' farming; and 3) farmers' responses to CAP.

(2) Soil data. Data on typical profiles of cinnamon soil were derived from self-collected field data, supplemented by references to soil census resources, such as <Aohan banner soil resource inventory>, <Inner Mongolia soil resource inventory> and <Cropland and scientific fertiliser application in Aohan banner>. Legacy soil map data were obtained from the Third national soil census at a 1:50,000 scale.

(3) Elevation data. ALOS 12.5m DEM data, Digital Elevation Model (DEM) data acquired by the Japan Aerospace Agency (JAXA) via the ALOS (Advanced Land Observing Satellite) satellite. ALOS 12.5m DEM data are available through NASA's Earthdata platform

(4) Soil erosion intensity. To assess the impact of soil erosion on CAP, this study incorporates soil erosion intensity as a variable in the model. According to the "Soil Erosion Classification and Grading Standards," soil erosion is classified into hydraulic, wind, and freeze-thaw erosion types. Erosion intensity is further categorized into slight, moderate, strong, and extreme based on the average erosion modulus (t/km²×a: <1000, 2500, 5000, 8000, 15000) and the average loss thickness (mm/a: <0.74, 1.9, 3.7, 5.9, 11.1). This data can be found in ([www.resdc.cn](www.resdc.cn)) for free.

### Ethics statement

This study did not require ethics approval because it involved non-invasive field surveys conducted at the village level. Upon arriving at each village, we first contacted the village secretary or leader to explain the purpose of our research and to gather basic information about the village. After obtaining their permission, we proceeded to interact with individual farmers. Prior to conducting the survey and completing questionnaires, we identified ourselves to the farmers and explained the purpose of the survey, the intended use of the data, and the confidentiality measures in place. Verbal informed consent was obtained from each participant before proceeding with the survey. No personally identifiable information was collected during the study, and all data were anonymized before analysis to ensure participant privacy.

### Field permit

No formal fieldwork permit was required for this study. The field surveys were conducted in publicly accessible rural areas that do not fall under any special administrative, ecological, or protected zones. Prior to administering questionnaires in

each village, the research team visited the village leaders (typically the local Party Secretary or village committee head) to explain the purpose of the study and to obtain verbal consent for conducting the survey within the community. This local engagement ensured community support and ethical conduct of fieldwork activities. All data were collected in accordance with local regulations and institutional guidelines.

## Results

### Identifying CAP: Evidence from the survey

A survey was conducted in several townships of Aohan Banner to collect data for comprehending and identifying CAP. Based on the questionnaire responses, 79 percent of the 33 villages reported the presence of CAP, while no CAP were found in the remaining villages. Of the 90 valid farmer questionnaires, 58% of the farmers confirmed the presence of CAP in their croplands and provided a series of generalized descriptions of CAP guided by the researcher. In this paper, these descriptions are summarized into a number of categories and the results are shown in Table 5.

Of the cropland with CAP, 98% were classified as arid land (Table 5A). Additionally, 88% of CAP occurrences were observed in the form of lumps, which generally refers to carbonate nodules (Table 5B). The majority of farmers reported that yields were lower in croplands affected by CAP compared to unaffected croplands (Table 5D). Further, among the 43 farmers who perceived reduced yields due to CAP, their main concerns are summarized in (Table 5E). Farmers' responses to CAP varied. The survey showed that 76% of farmers with CAP-affected cropland had not taken targeted measures to address the issue. In contrast, 18% of the farmers practiced deep tillage and manually remove carbonate lumps from the surface and deeper soil layers (Table 5F). These findings highlight the widespread presence of CAP, its detrimental impact on crop yields, and the limited farmer response, providing both a practical context and theoretical foundation for the study.

Furthermore, for cropland where CAP was present, the location of these cropland and the surrounding topographical conditions were recorded in detail (Table 5C). Overall, 69% of CAP-affected croplands were located on (gentle) slopes, 21% on flat land, 8% on terraces, and 2% on peak. These findings suggest that CAP is more likely to occur on sloping terrains, indicating that topographical factors play a significant role in CAP development. In addition to providing supplementary data to the construction of machine learning models, the findings of the survey provide a realistic basis for the interpretability of modelling results.

### Comparison of model performance

The relevant accuracy metrics were calculated by fitting various models (Table 6). In terms of AUC and LL, both BRT and XGBoost performed the best, while SVM performs poorly in the LL test, and LR underperformed in both AUC and LL. Further, we test the overall accuracy of these models under different thresholds. LR remained the poorest performer. Under the 60% threshold, the differences between BRT, XGboost and SVM were minimal, with all achieving an OA greater than 0.7. As the threshold increased, the OA for all models decreased significantly. In general, the tree-based models (BRT and XGBoost) demonstrated better performance compared to SVM and LR. In a side-by-side comparison, when the AUC is greater than 0.7 [49], it indicates good model accuracy. It shows that the tree model is relatively suitable for dealing with the classification of the CAP.

To compare the mapping ability of these models, the spatial distribution of CAP on cropland was predicted using the previously constructed model (Fig 4). There were notable differences in mapping ability between models. The predictions of the XGBoost and SVM models are overall large, while the predictions of the LR model are low. The BRT model, on the other hand, produced a more balanced representation of CAP distribution across the landscape.Considering both accuracy and mapping performance, the BRT model showed significant potential. Therefore, we will conduct a more in-depth analysis using the relatively accurate results from the BRT.

**Table 5. Questionnaire survey on various issues and their statistical data.**

| Questions | Issues | Number | Percent |
|---|---|---|---|
| A.CAP-affected cropland type | Arid land | 50 | 98% |
| | Irrigated land | 1 | 2% |
| B.CAP general forms | Lumpy | 45 | 88% |
| | Laminated | 3 | 6% |
| | Dotted | 3 | 6% |
| C.CAP-affected cropland terrain | Flat | 11 | 13% |
| | Slope | 35 | 39% |
| | Terrace | 4 | 5% |
| | Peak | 1 | 1% |
| D.Yield differences between CAP-affected and unaffected cropland | CAP-affected cropland is less productive | 43 | 84% |
| | No significant difference | 7 | 15% |
| | CAP-affected cropland is more productive | 1 | 1% |
| E.Impact of CAP on cropland | Obstruction of field operations | 22 | 51% |
| | Poor water preserving capacity | 9 | 21% |
| | Thin cultivated soil layer | 6 | 14% |
| | No effect in normal years, but drought affects it more than other croplands | 6 | 14% |
| F.Farmers' responses to CAP | No targeted measures to address the issue | 39 | 76% |
| | Deep tillage and eliminate CAP by hand | 9 | 18% |
| | Deep tillage combined with organic fertilizer application | 1 | 2% |
| | Topsoil replacement | 1 | 2% |
| | One-year fallow | 1 | 2% |

### Identification and analysis of the spatial distribution of CAP

The model's accuracy varies under different threshold conditions (Table 6), suggesting that selecting the optimal threshold can improve mapping accuracy. By setting the recall at 0.8, the resulting Precision-Recall curve (Fig 5) indicates a precision of 0.75 at this recall level, corresponding to a threshold of 0.62.

Based on the optimal thresholds obtained from the Precision-Recall curve, a binary classification map was generated (Fig 6). Under this threshold, areas with CAP are less likely to be missed. Black and red triangles indicate CAP presence or absence at their respective geographic locations.

Based on the optimal threshold, the portion of the probability value less than 0.62 is considered as the absence of CAP, which is indicated in orange; The portion of the probability value greater than 0.62 is considered as the presence of CAP, which is indicated in blue. Locations with a probability value below 0.62 are considered to have no CAP, marked in orange, while remained are classified as having CAP, marked in blue. Using the actual cinnamon soil points for validation (S1 Table), the model achieved an empirical accuracy of 75% based on the formula. In terms of CAP distribution, cinnamon soil cropland in the eastern and central regions is more prone to CAP; whereas areas along "gullies" are less likely to exhibit CAP. These trends are also reflected in the predictions from different models (Fig 4).

To further analyze the spatial distribution of CAP, a hotspot analysis was conducted based on the BRT prediction results (Fig 7a). Hotspots indicate areas with significantly high probability of occurrence of CAP, while coldspots indicate

**Table 6. Accuracy performance of different models.**

| Models | AUC | LL | OA(60%) | OA(70%) | OA(80%) |
|---|---|---|---|---|---|
| BRT | 0.721 | 0.559 | 0.706 | 0.650 | 0.472 |
| XGboost | 0.707 | 0.560 | 0.705 | 0.642 | 0.514 |
| SVM | 0.572 | 0.587 | 0.705 | 0.662 | 0.322 |
| LR | 0.551 | 0.623 | 0.667 | 0.469 | 0.321 |

areas with significantly low probability, both of which passed the significance test ($p < 0.01$). The hotspot map reinforces the findings from (Fig 6), confirming that CAP occurrence is less probable in the croplands along the "gullies".

This trend is clearly influenced by topography. To explore this further, we analyzed the result using the Geo variable (Fig 7b), which shows the distribution of cold hotspot regions based on different terrain types (Fig 7c). Overall, the area of the hotspot region (800.6km$^2$) was significantly larger than that of the coldspot region (471.6km$^2$). The area of cropland on slope is about 576km$^2$, which is significantly higher than that of other terrains. To quantify the spatial distribution trends, the ratio of the cold-hot zone area (RCH) was calculated (Fig 7c). The overall RCH is 0.589; in water-dispersed terrains, such as slopes, spurs, shoulders, ridges and peaks, the RCH was 0.432. In contrast, in the sedimentary terrains like pits, footslopes, valleys and hollows, the RCH was 1.059. This suggests that cold zones are more prevalent in sedimentary terrains compared to hot zones. On a terrain-by-terrain basis, RCH showed a gradual increase ($p < 0.05$) starting from the peak terrain. This suggests that as the terrain changes from dry to wet, the proportion of cold areas increases, which means that the probability of CAP decreases. These findings suggest that CAP is less likely to occur in depositional terrains and more likely in moisture-drying, dispersive terrains.

## Characterization of topographic features related to CAP

Among the various covariates (S1 Fig), MRVBF had the highest contribution, responding to the core driver of topographic influence on the formation of the CAP. To further analyze the relationship between topography and CAP, a scatter plot of MRVBF versus CAP probability was plotted (Fig 8). Darker coloured squares in the graph indicate a higher density of scatter at that location. The scatter density trend shows that the probability of CAP decreases as MRVBF values increases. A linear correlation was fitted to show a significant negative correlation between MRVBF and the probability of CAP ($p < 0.05$). MRVBF effectively distinguishes between hillslopes and valley floors, which in turn separates erosional from depositional areas [40]. High values of MRVBF correspond to flatter terrain, which are usually associated with river valleys, while lower values represent areas of steeper or uneven terrain. The negative correlation suggests that in depositional terrains like river valleys, CAP is less likely to occur (probability<0.15), whereas in areas of steep or erosion, the likelihood of CAP occurrence is higher(probability>0.75).

Such a pattern has a similar representation in the Geo variable (Fig 9). The 3D morphology corresponding to different values is shown in (Fig 9c), with terrain 1 corresponding to a Geo value of 1, and so on [41]. Topographies 2, 3, 4, and 5 are characterized by higher elevations around the central point, indicating erosive conditions with moisture dispersion and drying. On the contrary, topographies 7, 8, 9, and 10 have lower surrounding elevations, indicating depositional conditions with moisture accumulation and increased wetness. This transition from dry to wet terrains is captured numerically through Geo values.

In (Fig 9b), the violin plot is used to represent the distribution of CAP probability across different Geo values. Overall, the probability density of CAP tends to appear in the lower part of the violin plot while Geo values increased. This indicates that the probability tends to decrease as the Geo value increases. Linear regression further confirms a significant negative correlation between Geo and CAP probability ($p < 0.05$).

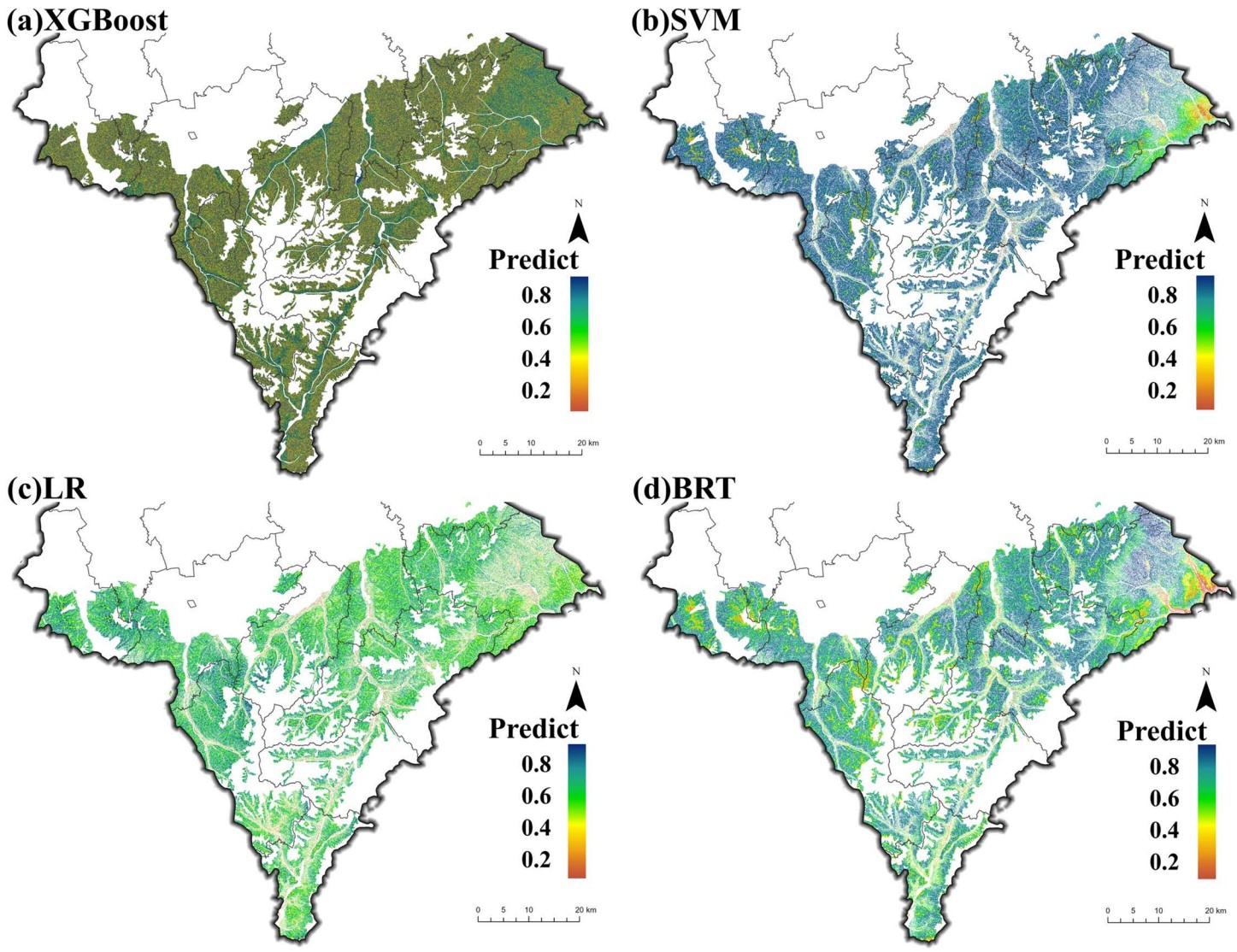

**Fig 4. Spatial distribution of CAP occurrence probabilities predicted by different models.**

Geo values follow a regular progression from 1 to 10 (Fig 9c), representing a transition from drier to wetter terrain conditions. The apparent negative trend in (Fig 8b) implies that as terrain moisture becomes more aggregated and wetter, the CAP probability decreases. Conversely, in drier, moisture-dispersed terrains, CAP probability increases. Further global statistics support this trend, showing a higher probability of CAP in moisture-dispersed terrains (72%) compared to moisture-aggregated terrains (58%). Relatively more water infiltration reduces the probability of CAP.

## Discussions

### Modelling based on LSM is feasible

Overall, machine learning models can capture non-linear relationships in soil assessment and have strong generalisation capabilities. It also allows for theoretical analyses through model interpretation. And, it has considerable advantages for scaled tasks.

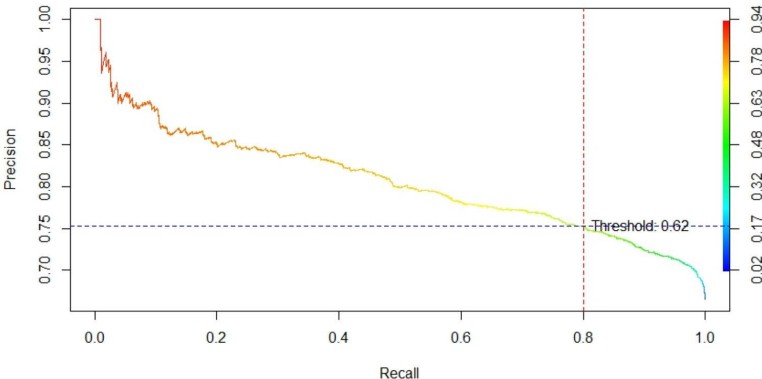

**Fig 5. Optimal recall curve.**

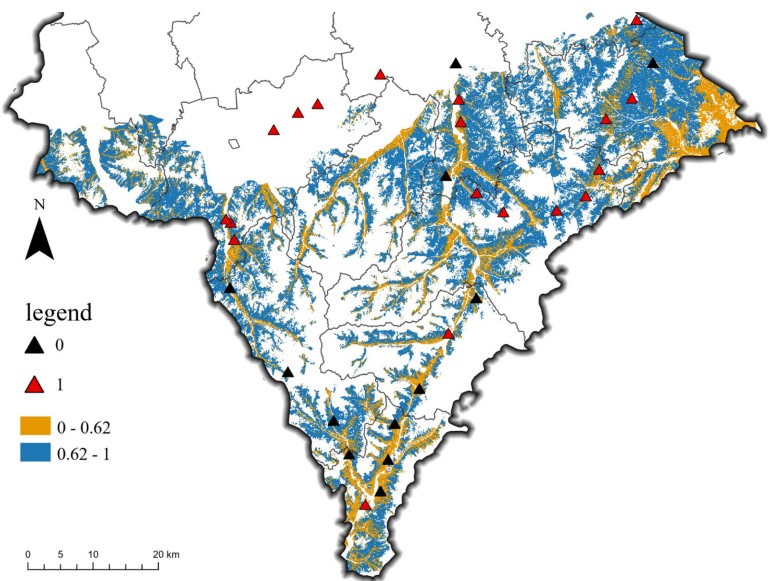

**Fig 6. CAP spatial distribution based on optimal threshold.**

This article constructs a dataset based on LSM and utilizes multiple machine learning models for modeling. The accuracy of different models for predicting CAP was evaluated. The results highlight that tree models, particularly BRT, perform well in both accuracy and mapping capability. Using LSM for soil mapping is a feasible and important method, especially in areas where survey costs are prohibitively high [50]. By leveraging LSM, soil mapping efforts can be expedited, reducing the time and cost associated with traditional soil sampling and field surveys [51,52].

The dataset sampling from LSM contains a wealth of soil attribute information and spans a broad geographical area, making it especially beneficial for regions with limited access to extensive field data. However, one of the key challenges is the insufficient number of soil profile points available for model construction in Aohan Banner. To address this, 10,000 random sampling points were selected from the LSM to build a comprehensive sample dataset. While this large sample size helps maximize data utility, some misclassified or false sample points are inevitable. Despite this, the BRT model still demonstrates an acceptable level of accuracy, showcasing the potential and feasibility of using LSM-based data for soil mapping.

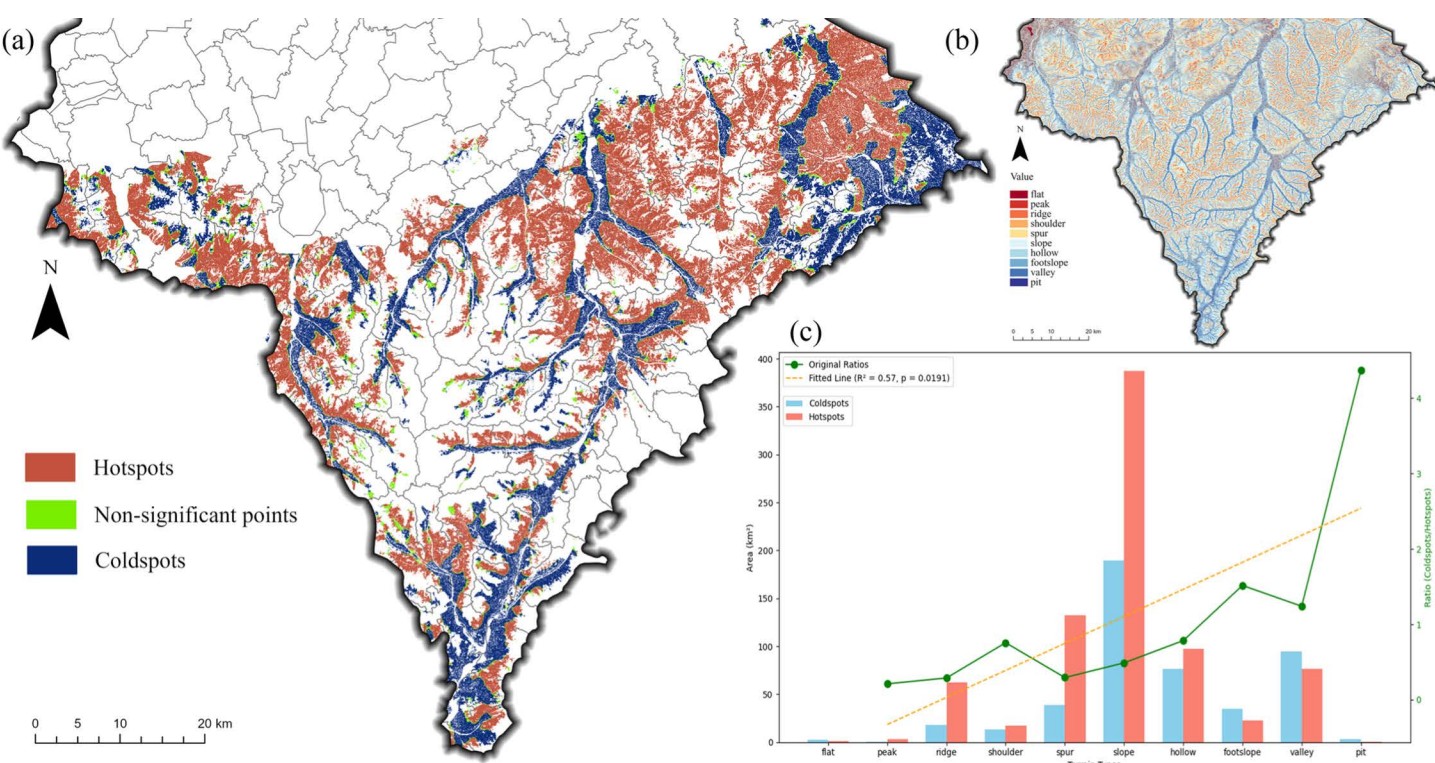

**Fig 7. Hotspot analysis of CAP. (a)** Hotspot map of CAP occurrence probability; **(b)** Spatial distribution of Geo variable; **(c)**Area distribution of hotspots with CAP occurrence probability in different terrains.

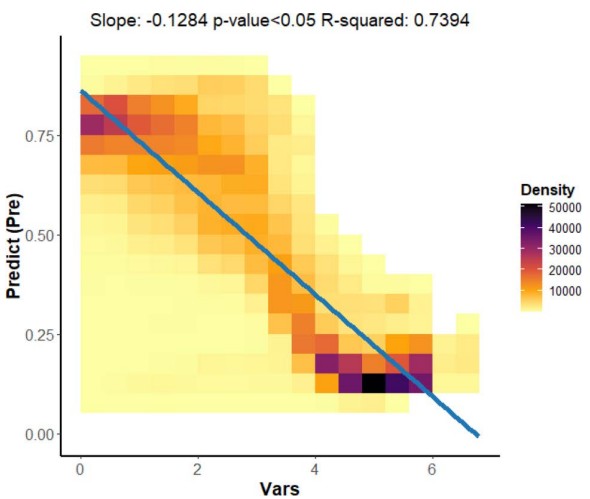

**Fig 8. Scattered trend plot of MRVBF.**

Among the models tested, tree-based methods consistently performed better in terms of prediction accuracy (Table 4). Specifically, the BRT model outperformed XGBoost in both accuracy and mapping performance. One reason for this may lie in the tree model's ability to handle missing data and capture nonlinear relationships more effectively [53]. While

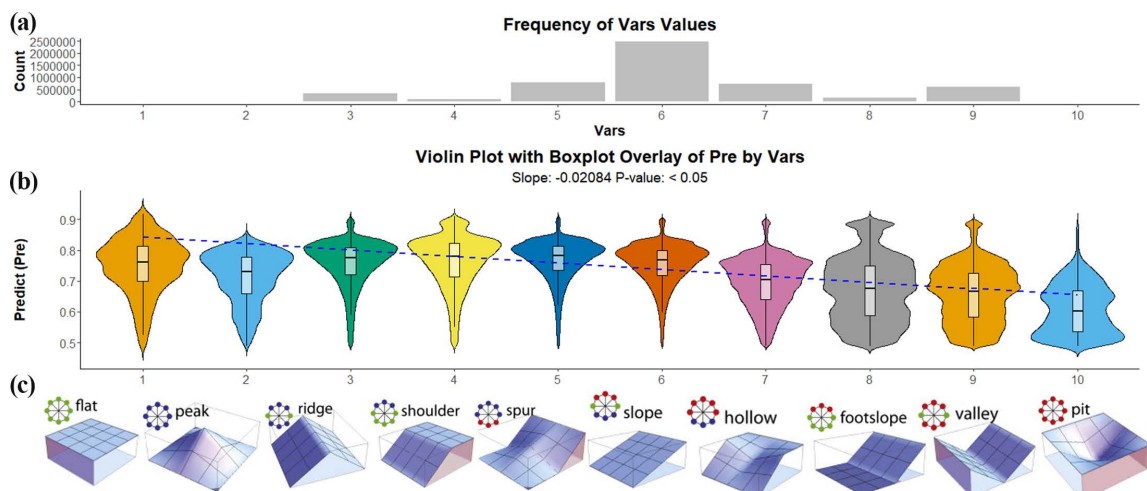

**Fig 9. Different Geo values correspond to the 3D morphology of the terrain [ 41 ]. (a)**Frequency of Geo values; (b)violin plot with boxplot overlay of CAP probability in Geo; **(c)**The 3D morphology of Geo.

XGBoost has a deeper and more complex tree structure, this can sometimes lead to drastic changes and extreme predictions, particularly in spatial mapping [46]. In contrast, the shallower tree structure of BRT results in smoother predictions and more reliable spatial outcomes, making it better suited for this type of mapping task [45].

However, the modeling process in this study primarily focuses on topographic factors. As a result, deviations in predictions may occur when other influential factors are not considered. For example, in agricultural systems, irrigation and fertilization increase the respiration of roots and microorganisms, which promotes the formation of PC [54]. Soil moisture conditions, which fluctuate throughout the growing season, also affect the dissolution, movement, and formation of PC [18]. Continuous irrigation can also influence the vertical movement of PC, leaching it from the shallow soil into deeper layers. These environmental factors, not accounted for in the current model, may contribute to the observed discrepancies and misclassifications in the predicted results.

## CAP is more likely to occur in terrains at higher risk of soil erosion

By analyzing the relationship between prediction results and topographic variables, we found that CAP is more likely to occur in topographies characterized by moisture divergence and dryness. The essence of CAP is the accumulation of PC, and climate is considered the main controlling factor for the formation of PC [5,7]. However, at smaller scales, where climate variation is negligible, topographic conditions emerge as the primary determinants of CAP distribution. Topographic conditions influence the accumulation of PC by affecting the distribution of water and heat [7].

Similar patterns are evident in studies from other regions [3,9,20,55]. Dunling wang initiated a study on the landscape distribution of PC in chernozemic soil in the Southeastern Saskatchewan region. Whose results indicated that strongly sedimentary topographic environments, such as slopes and depressions, facilitate the removal of shallow PC [20]. This occurs because runoff from adjacent slopes creates wetter soil conditions compared to upslope areas, promoting PC leaching in the shallow layers [56]. G. Jacks studied the relationship between calcareous horizons and hillside landscapes in southern India [8]. The results show that, due to the movement of carbonate solutions, it is more likely to form calcrete at deeper levels. Additionally, Some studies in Mexico have indicated that soils on both sides of ridges, slopes, or in runoff positions have shallower PC accumulation due to reduced water infiltration [7,10]. Such shallow accumulation of PC is closely associated with CAP formation.

The findings of this study align with these observations. We found that areas with a lower probability of CAP often correspond to sedimentary or moist topographic conditions, while areas with a higher probability of CAP correspond to divergent or dry topographic conditions (Fig 7-9). Dry or divergent terrain conditions often indicate an increase in runoff speed and water erosion potential, leading to a sharp rise in the risk of soil erosion. This suggests that CAP in cinnamon soil cropland is more likely to occur in areas with higher soil erosion risk.

For this extended view, we introduce an external dataset to validate it. As the intensity of soil erosion increases, the probability of CAP also gradually rises (Fig 10.). The analysis reveals that as the erosion intensity increases, the probability of CAP becomes more stable and higher. Although extreme erosion has fewer samples, making its statistical representation less robust, areas with slight erosion consistently exhibit lower probabilities of CAP. This trend aligns with the study's main findings, suggesting that an increase in erosion intensity leads to a higher probability of CAP.

These terrains with higher risk of soil erosion, weakening the downward infiltration of moisture, affecting the leaching of PC in the soil. On the other hand, these terrains are often accompanied by the erosion of surface soil. This may lead to the relative upward movement of the original PC accumulation layer, thereby causing the occurrence of CAP.

## Recommendations for mitigation strategies

Combined with field research and analysis of results, we believe that long-term irrigation and deep plowing may be effective mitigation strategies that contribute to sustainable soil management. Under uniform irrigation conditions, the increase of PC in the 100–160 cm soil layer accounts for 70%−90% of the total, while the increase in the 0–100 cm soil layer is relatively small [57]. Related studies [17] have found that at the end of the dry season, the PC reserves at the top 30 cm of soil are, on average, 15% higher than at the end of the rainy season. Long-term irrigation experiments indicate that irrigation can effectively leach PC to depths of 1–4 meters [19,58].

These findings suggest that irrigation is a favorable method for reducing PC in the shallow layers of cropland (0–100 cm). In practical surveys, we found that irrigation is indeed a useful method: there were almost no reports of CAP in irrigated fields (Table 5A); whereas in dry years, the lack of irrigation is a major problem for reduced yields in dryland farming. And when the average annual rainfall is less than 500 mm, it is common for PC to accumulate on the soil surface [21].

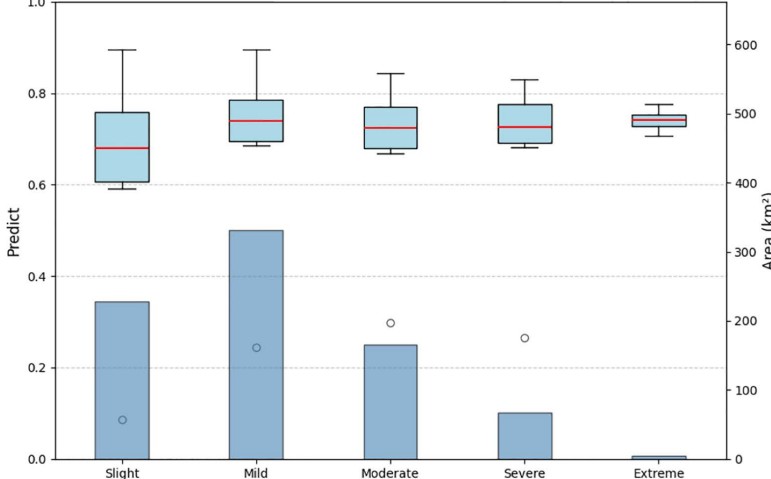

**Fig 10. Relationship between predicted values and soil water erosion intensity with area distribution.** The bar chart represents the area distribution of different soil erosion intensities in cinnamon soil cropland; the box-and-line plot represents the CAP probability distribution of different soil erosion intensities in cinnamon soil cropland.

Although there are limited articles on how deep tillage affects the formation of pedogenic carbonates (PC) [11], its impact on PC cannot be ignored. Due to the semi-arid and semi-humid location of Aohan Banner, water resources are scarce. The accumulation of PC in the shallow layers blocks soil pores and may also create a certain degree of root restriction layer [3]. Deep tillage can break through the root restriction layer in the soil [59]. This helps to create more pore space in the soil [60], reduce water runoff [61], increase water infiltration [62], decrease water evaporation [63], while also suppressing weed growth and alleviating the problem of soil and water loss in cropland. Additionally, the cracks created by deep tillage in the soil help promote the downward movement of carbonates [11], indirectly affecting the leaching of surface PC.

Deep tillage includes various methods, primarily deep plowing and deep loosening in Aohan banner. For PC with shallow sediment depth, blind deep plowing may cause adverse effects. Therefore, deep plowing is often combined with the application of organic fertilizers or amendments to achieve better results [59]. Thus, the depth of deep plowing needs to take into account the sediment depth and strength of PC to prevent mixing of its fragments with the topsoil. In contrast, deep loosening can be a successful method for addressing similar issues [64,65]. This is because deep loosening can break the root restriction layer without turning the soil. This prevents the mixing of PC fragments with the topsoil while also improving the physical properties of the soil.

In summary, irrigation and deep tillage may be effective methods to address CAP in cinnamon soil cropland. While leaching the shallow soil carbonates, these methods can also improve the physical and chemical properties of the soil and enhance the plow layer. Local implementation of solutions should be based on actual conditions, such as the availability of external water sources, the depth of PC accumulation, and the topographical conditions of the cropland. Based on these conditions, the management characteristics of different regions should be clarified to facilitate further field experiments to determine the practical effects of these methods.

## Limitations

Our paper creates a framework for identifying CAP on cropland, using LSM and combining both survey and machine learning methods. It enables governments and managers to optimise land-use planning and to implement differentiated management policies. The framework provides a theoretical basis for the sustainable management of local soils.

However, there are some limitations to our study. To a large extent, the LSM is our basic data. The accuracy of the LSM and the level of detail of the information available are related to the precision of the identification results. To date, the results of the third national soil census have not been widely disseminated, and most of the soil data still remain from the second soil census. This leads to the lack of accuracy of the results. The results of the third soil census are more detailed and their distribution is more accurate with the addition of emerging technologies. Therefore, in the future, when the third soil census results are popularized, it can effectively improve the accuracy of the results in the framework of this paper. Aohan banner is an extended area of Liaohe basin. Most of all, Aohan banner is the typical area in North cinnamon soil zone. But, Limited by the difficulty of obtaining high-precision soil data over a wide area, we have only conducted research in Aohan banner, with a view to providing a reference for other regions.

## Conclusion

In this paper, a novel approach to constructing datasets is used, i.e., LSM sampling data rather than relying on traditional survey data. This approach improves data accessibility and provides a low-cost solution for soil characterization data construction for regions lacking profile data. Moreover, the relationship between CAP and terrain features was revealed by an effective machine learning model.

The results show that tree-based models, particularly the Boosted Regression Trees (BRT), provide higher accuracy and better mapping performance compared to XGBoost. This approach to constructing soil characteristic datasets demonstrates significant feasibility, offering a cost-effective alternative for regions with limited soil profile data.

The predictive results indicate that cropland along the "gullies" in Aohan Banner is less prone to CAP formation. Notably, the ratio of cold to hot zones increases as the terrain transitions from dry to wet ($p < 0.05$). Moreover, the global analysis of topographic variables reveals that CAP is more likely to occur in dry, divergent terrains ($p < 0.05$), which are also associated with higher risks of soil erosion. These terrains hinder moisture infiltration, promoting the accumulation of calcium in the shallow soil layers. Additionally, areas with high soil erosion risk may exacerbate CAP through the erosion of topsoil, which brings the calcium-rich layers closer to the surface.

Survey results underscore the widespread nature of CAP and its significant impact on agricultural productivity. However, farmers are generally unaware of effective mitigation strategies. Irrigation and deep tillage are suggested as potentially effective measures to mitigate CAP formation.

## Supporting information

**S1 Fig. Contribution of covariates in different models.**
(TIF)

**S1 Table. Actual soil profile data.**
(XLSX)

## Acknowledgments

We would like to thank the Aohan Banner Government for their assistance with the questionnaire and data collection.

## Author contributions

**Data curation:** Xingjie Yin, Pingli An.

**Formal analysis:** Xingjie Yin.

**Funding acquisition:** Pingli An.

**Investigation:** Xingjie Yin, Haile Zhao, Yuling Jin, Wenting Liu.

**Methodology:** Xingjie Yin, Haile Zhao, Zhihua Pan, Pingli An.

**Software:** Xingjie Yin.

**Supervision:** Zhihua Pan, Pingli An.

**Validation:** Xingjie Yin.

**Visualization:** Xingjie Yin.

**Writing – original draft:** Xingjie Yin, Haile Zhao, Yuchao Luo, Yuling Jin, Pingli An.

**Writing – review & editing:** Xingjie Yin, Haile Zhao, Yuchao Luo, Yuling Jin, Zhihua Pan, Pingli An.

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
