## [Decision Letter · Decision Letter 0]

10 Mar 2025

PONE-D-25-00681A framework for identifying calcium accumulation problem in cropland: Integrating field surveys, legacy soil map, and machine learning modelsPLOS ONE

Dear Dr. An,

Thank you for submitting your manuscript to PLOS ONE. After careful consideration, we feel that it has merit but does not fully meet PLOS ONE’s publication criteria as it currently stands. Therefore, we invite you to submit a revised version of the manuscript that addresses the points raised during the review process.

We look forward to receiving your revised manuscript.

Kind regards,

Vivek Sivakumar, Ph.D

Academic Editor

PLOS ONE

Journal Requirements:

2. You indicated that ethical approval was not necessary for your study. We understand that the framework for ethical oversight requirements for studies of this type may differ depending on the setting and we would appreciate some further clarification regarding your research. Could you please provide further details on why your study is exempt from the need for approval and confirmation from your institutional review board or research ethics committee (e.g., in the form of a letter or email correspondence) that ethics review was not necessary for this study? Please include a copy of the correspondence as an ""Other"" file.

4. In your Methods section, please provide additional information regarding the permits you obtained for the work. Please ensure you have included the full name of the authority that approved the field site access and, if no permits were required, a brief statement explaining why.

5. Thank you for stating the following financial disclosure: [This work was supported by the National Key Research and Development Plan of China (2022YFD1500602-3) and the National Natural Science Foundation of China (grant number 42271268)]. 

6. We note that you have indicated that there are restrictions to data sharing for this study. For studies involving human research participant data or other sensitive data, we encourage authors to share de-identified or anonymized data. However, when data cannot be publicly shared for ethical reasons, we allow authors to make their data sets available upon request. For information on unacceptable data access restrictions, please see http://journals.plos.org/plosone/s/data-availability#loc-unacceptable-data-access-restrictions.

7. Your ethics statement should only appear in the Methods section of your manuscript. If your ethics statement is written in any section besides the Methods, please move it to the Methods section and delete it from any other section. Please ensure that your ethics statement is included in your manuscript, as the ethics statement entered into the online submission form will not be published alongside your manuscript.

8. We note that Figures 1,3,4,5, and 6 in your submission contain [map/satellite] images which may be copyrighted. All PLOS content is published under the Creative Commons Attribution License (CC BY 4.0), which means that the manuscript, images, and Supporting Information files will be freely available online, and any third party is permitted to access, download, copy, distribute, and use these materials in any way, even commercially, with proper attribution. For these reasons, we cannot publish previously copyrighted maps or satellite images created using proprietary data, such as Google software (Google Maps, Street View, and Earth). For more information, see our copyright guidelines: http://journals.plos.org/plosone/s/licenses-and-copyright.

1. You may seek permission from the original copyright holder of Figures 1,3,4,5, and 6 to publish the content specifically under the CC BY 4.0 license. 

“I request permission for the open-access journal PLOS ONE to publish XXX under the Creative Commons Attribution License (CCAL) CC BY 4.0 (http://creativecommons.org/licenses/by/4.0/). Please be aware that this license allows unrestricted use and distribution, even commercially, by third parties. Please reply and provide explicit written permission to publish XXX under a CC BY license and complete the attached form.

In the figure caption of the copyrighted figure, please include the following text: “Reprinted from [ref] under a CC BY license, with permission from [name of publisher], original copyright [original copyright year].

Additional Editor Comments:

Dear Author,

The manuscript has been written well but since some improvements are required kindly include all the statements mentioned in the revised manuscript.

Thank You.

Reviewers' comments:

Reviewer's Responses to Questions

**Comments to the Author**

1. Is the manuscript technically sound, and do the data support the conclusions?

Reviewer #1: Yes

Reviewer #2: Yes

2. Has the statistical analysis been performed appropriately and rigorously? 

Reviewer #1: Yes

Reviewer #2: Yes

3. Have the authors made all data underlying the findings in their manuscript fully available?

Reviewer #1: Yes

Reviewer #2: Yes

4. Is the manuscript presented in an intelligible fashion and written in standard English?

Reviewer #1: Yes

Reviewer #2: Yes

5. Review Comments to the Author

Reviewer #1: What are the key factors contributing to calcium accumulation in croplands?

How does the study integrate field surveys with machine learning models?

What role do legacy soil maps play in identifying calcium accumulation?

Which machine learning models were used in the framework?

What soil properties were considered in the study?

How does calcium accumulation affect crop productivity?

What were the major data sources for the analysis?

What are the advantages of using machine learning for soil assessment?

What were the key findings of the study?

How can this framework help in sustainable soil management?

Reviewer #2: 1. Lack of Novelty - Novelty of the Manuscript is missing

2. The selection of machine learning models (BRT, XGBoost, SVM, and LR) was not well justified.

3. The study was conducted in a specific region (Aohan Banner, Chifeng City) - Compare with any other region

4. Suggested to include additional evaluation metrics and external validation datasets to strengthen the conclusions.

5. It is noted that the discussion section did not adequately acknowledge the limitations of the study

6. This manuscript does not provide strong recommendations for practical mitigation strategies

7. Some sections of the manuscript were noted as difficult to follow due to unclear phrasing and grammatical errors - Change it

6. PLOS authors have the option to publish the peer review history of their article (what does this mean? ). If published, this will include your full peer review and any attached files.

**Do you want your identity to be public for this peer review?** For information about this choice, including consent withdrawal, please see our Privacy Policy .

Reviewer #1: **Yes: ** Priya

Reviewer #2: No

---

## [Author Response · Author response to Decision Letter 1]

15 Apr 2025

Dear Editors,

We appreciate the opportunity to revise our manuscript [A framework for identifying calcium accumulation problem in cropland: Integrating field surveys, legacy soil map, and machine learning models] and would like to sincerely thank you for your constructive feedback. Below, we provide a detailed response to each of the editorial comments and outline the revisions or explanations accordingly.

1. Ethics Approval

We confirm that our study does not require ethics approval, as it does not involve human or animal subjects, and no personal or sensitive information was collected. To comply with the editorial request, we have obtained a formal explanation letter from our affiliated institution—College of Land Science and Technology, China Agricultural University.

The statement was reviewed and approved by:

1) Dr. Haishuang Jia, Director of Scientific Research and Social Services,

2) Ms. Zhaohong Cao, Deputy Party Secretary and Vice Dean,

3) Ms. Xiaohe Yu, Chief Officer of General Affairs.

We have uploaded the scanned letter, including the official stamp, in both Chinese and English versions, under “Other” files for your reference.

2. Data Availability

We regret that we are unable to fully share our key dataset due to confidentiality agreements and protection of residents' geospatial privacy. These restrictions are in compliance with local data governance regulations.

However, all openly available data sources used in the study have been listed in the manuscript. Further explanation regarding the data sharing constraints and the contact information for data requests has been added to the Data Availability statement.

3. Code Availability

We respectfully clarify that our modeling code is not fully original and was developed based on the widely referenced method by Elith et al. As such, it is not applicable to publish this code as a standalone contribution.

Furthermore, the key modeling parameters, data processing steps, and algorithm settings have been clearly outlined in Table 4 of the manuscript for full transparency. We believe this provides sufficient reproducibility for readers. Should any issues arise, we remain available for clarification.

4. Map and Satellite Image Copyright

We have addressed the copyright concerns in the following ways:

1) Figure 1: The original basemap has been removed.

2) Figures 1, 3, 4, 5, and 6: All geographic boundary data (including China’s administrative boundaries) were obtained from the National Geomatics Center of China, a publicly accessible and free source.

We have updated the captions and methods section to reflect the data source and have included the map approval number GS(2024)0650 in the manuscript.

5. Field Permits and Ethics Statement Placement

We have revised the Methods section to clarify that no field permit was required, as the study area is public and no environmental disturbance or restricted activity was involved.

6. Funding Statement

As requested, we have added the following standard disclaimer to the “Acknowledgments” of the manuscript:“The funders had no role in study design, data collection and analysis, decision to publish, or preparation of the manuscript.”

7. File Uploads and Formatting

We have prepared and uploaded the following revised materials according to PLOS ONE requirements:

1) Response to Reviewers (this document)

2) Revised Manuscript with Track Changes

3) Clean Manuscript (without track changes)

4) Ethics Statement Document and Data availability (uploaded under “Other” files)

We hope these revisions and clarifications adequately address the editorial concerns. Please feel free to contact us if further information is required.

Sincerely,

[Pingli An]

[College of Land Science and Technology, China Agricultural University]

Email: [anpl@cau.edu.cn]

**Reviewer#1**

1. What are the key factors contributing to calcium accumulation in croplands?

Thank you very much for your insightful question. Identifying the key factors contributing to calcium accumulation is indeed crucial and deserves a clear explanation in both the Introduction and Methods sections. Your comment prompted us to revisit and revise the relevant content accordingly.

Calcium accumulation problems (CAP) in soils primarily arise from the excessive buildup of pedogenic carbonates (PC). Therefore, understanding the drivers of PC accumulation is essential for addressing the mechanisms behind CAP. Previous studies have extensively examined PC formation, revealing that climate, topography, and human activities are the dominant influencing factors. Under the influence of these drivers, PC undergoes repeated processes of dissolution, translocation, and re-precipitation, eventually forming various morphological features in the soil.

In the context of our study, CAP is also controlled by these primary factors—climate, topography, and human activities. Given the specific characteristics of the region under investigation, we further emphasize the role of parent material, which provides a sufficient source of calcium carbonate for accumulation. While climate sets the general environmental background, its variation is relatively limited within the study area. Human activity, although important, is constrained by data availability. Therefore, we focus on topographic factors, which strongly influence moisture distribution at the soil surface under similar climatic conditions. These micro-environmental differences play a critical role in the formation and spatial differentiation of CAP. The detailed descriptions are as follows:

Line 49-57: “PC is deposited in various forms (including earthworm biospheroliths, rhizoliths, pseudomycels, nodules, coatings and calcrete) through the dissolution, movement and re-precipitation of calcium carbonate[3]. PC is further redistributed under the influence of climate[4–6], topography[7–10], human activities[11–14], etc. A sediment layer of a certain depth is formed through eluviation and illuviation of PC, which varies with precipitation[11,15]. The amount of PC increases with the land use changes to cropland[11,13,16]and gradually accumulates in the shallow layers[17,18]. Irrigation can eluviate PC to deeper layers[17,19], while in arid regions without additional water inputs, PC tends to accumulate in the shallow layers of croplands[20,21].”

Line 102-107: “CAP is defined as a soil characteristic that negatively impacts crops by degrading the physical and chemical properties of the soil due to the accumulation of carbonates. PC is the primary component of these accumulated carbonates, with calcium carbonate being the main constituent of PC[3]. The essence of CAP lies in the accumulation of PC, therefore factors influencing PC formation are key contributors to CAP, like parent material, topography, climate and human activity, etc.”

2. How does the study integrate field surveys with machine learning models?

Thank you very much for this important question. The integration of field survey results with machine learning models is indeed one of the core aspects of our study. However, we acknowledge that this was not clearly articulated in the original manuscript. Thanks to your careful review, we have revised the relevant sections to better explain this integration.

In brief, the field survey component includes both questionnaire responses and soil sampling data. The questionnaire results helped validate our understanding of the CAP phenomenon and provided theoretical support for the construction of the machine learning models. They also offered contextual insights into local perceptions and experiences, which reinforced the rationale for selecting certain modelling variables. On the other hand, the soil sampling results were used to verify both the LSM (legacy soil map) and the predictions generated by the machine learning models. This validation process strengthened the reliability of our findings and enhanced the overall credibility of the study. The detailed descriptions are as follows:

Line 281-284: “These findings suggest that CAP is more likely to occur on sloping terrains, indicating that topographical factors play a significant role in CAP development. In addition to providing supplementary data to the construction of machine learning models, the findings of the survey provide a realistic basis for the interpretability of modelling results.”

3. What role do legacy soil maps play in identifying calcium accumulation?

Thank you for your valuable question. The use of legacy soil map (LSM) in identifying calcium accumulation problems (CAP) is a central element of our study. We initially addressed the role of LSM in the "LSM Sampling" section, but your insightful comment prompted us to review and revise this part to better emphasize their significance.

In our study, LSM serve as the foundation for constructing the CAP dataset. Specifically, LSM provide the “Y” variable—i.e., whether a CAP exists or not—and the spatial locations of sample points obtained through random sampling. These sampled locations were then used to extract relevant independent variables, enabling the construction of the full dataset for model training and analysis. The detailed descriptions are as follows:

Line 123-128: “The county-level LSM scale is 1:50,000, providing an accurate spatial distribution of various soil types for sampling. Before LSM sampling, we validated the soil map using data from 33 filed-dug soil profiles. (Fig 2) illustrates some of the profiles, with the phenomenon of CAP, forming points, blocks, and surfaces. For each profile, we recorded the actual observed soil type, coordinates, and CAP labeling information (S1 Table). The LSM was compared to the actual soil types, yielding an overall accuracy of 94%, confirming the reliability of the LSM.”

Line 131-135: “When "carbonate deposits" were observed in a typical soil profile, it is assumed that CAP was present in the corresponding soil type. (Fig 3a) illustrates the distribution of soil types in southern Aohan Banner, where 628 of 1106 cinnamon soil polygons exhibit CAP. The number of sampling points per cinnamon soil polygon was determined based on area as weight and randomly sampled. A total of 10,000 sample points were allocated, some of which are shown in (Fig 3b).”

4. Which machine learning models were used in the framework

Thank you for your insightful question. The application of machine learning models is a key component of our approach to identifying calcium accumulation problems (CAP). Model selection is crucial, as it directly affects the accuracy and robustness of the results. Although we briefly outlined our model choices in the “Model Selection” section, we acknowledge that our original explanation lacked clarity. In response to your comment, we have revised and expanded this section to provide a clearer rationale.

In selecting the models, we considered both linear and non-linear relationships among the terrain variables. To capture complex, non-linear interactions, we applied three widely used machine learning models with varying levels of complexity: XGBoost, Boosted Regression Trees (BRT), and Support Vector Machines (SVM). Additionally, we included logistic regression to account for potential linear relationships, providing a baseline for comparison.

This combination allowed us to explore how different models interpret the relationship between predictors and CAP occurrence, ensuring a more comprehensive and interpretable modeling framework. The detailed descriptions are as follows:

Line 169-172: “For the purpose of resolving linear or non-linear relationships between terrain variables, we chose three common models (XGBoost, BRT and SVM, with decreasing model complexity in that order) to account for non-linear relationships between terrain variables, and Logistic regression to account for linear relationships between terrain variables (Table 4).”

5. What soil properties were considered in the study

Thank you for your thoughtful question. Considering soil properties is indeed essential for identifying calcium accumulation problems (CAP). In our study, some soil properties were directly or indirectly taken into account, though we did not explicitly elaborate on them in the original manuscript. We appreciate your attention to this aspect, which prompted us to revise and supplement the relevant content.

Specifically, we considered key soil physical and chemical properties such as calcium carbonate content and soil pore size, as these characteristics influence the formation and accumulation of calcium in the soil. Among these, only calcium carbonate content was explicitly used in our framework to determine the presence of CAP, serving as a criterion in labeling the data.

However, due to the limited spatial variability and availability of other soil property data, we did not include them as independent variables in the machine learning models. Instead, we relied on topographic factors, which reflect the spatial heterogeneity of the landscape and indirectly capture variations in soil properties relevant to CAP. The detailed descriptions are as follows:

Line 158-162: “In summary, spatial dissimilarity of variables is key to identification. Thus, we only wish to consider the soil physical and chemical properties, such as calcium carbonate content, soil pore size, etc. These properties manifest themselves differently in different terrains. We therefore identified CAP by considering topographic factors to indirectly consider the combined expression of soil properties.”

6. How does calcium accumulation affect crop productivity

Thank you for this meaningful question. We briefly mentioned the impact of calcium accumulation problems (CAP) on crop productivity in the introduction; however, your comment prompted us to revisit and improve the clarity of our explanation.

CAP affects crop productivity by altering the soil's physical and chemical properties. On the physical side, excessive calcium carbonate accumulation can clog soil pores and reduce water infiltration, which limits the availability of water to plant roots. On the chemical side, high levels of carbonate can interfere with the absorption of essential micronutrients, thereby impairing plant growth and yield. The detailed descriptions are as follows:

Line 58-62: “Calcium accumulation problem (CAP) caused by excessive PC accumulation, which affects the physical, chemical and biological properties of the soil[22], and thus indirectly affects crop productivity. Such as clogging soil pores[23], weakening inter-root water infiltration and movement[24], reducing plant utilization of micronutrients[15], and threatening sustainable agricultural development.”

7. What were the major data sources for the analysis

Thank you for your insightful question. Data sources are indeed foundational to our study. We revisited the “Data” section and improved the clarity of the descriptions regarding the datasets used in our framework. The major data sources include: field survey data (questionnaires), the Third National Soil Survey of China, local soil archive records (soil monographs), and topographic data. Each of these datasets contributed to different parts of the model construction and validation process. We also supplemented additional details to clarify specific sources, such as the data used for assessing soil erosion intensity. The detailed descriptions are as follows:

Line 235-240: “Soil erosion intensity. To assess the impact of soil erosion on CAP, this study incorporates soil erosion intensity as a variable in the model. According to the "Soil Erosion Classification and Grading Standards," soil erosion is classified into hydraulic, wind, and freeze-thaw erosion types. Erosion intensity is further categorized into slight, moderate, strong, and extreme based on the average erosion modulus (t/km²×a: <1000, 2500, 5000, 8000, 15000) and the average loss thickness (mm/a: <0.74, 1.9, 3.7, 5.9, 11.1). This data can be found in www.resdc.cn for free.”

8. What are the advantages of using machine learning for soil assessment?

Thank you for your valuable question. This is indeed a critical aspect of our study, as the use of machine learning plays a central role in soil assessment and the identification of calcium accum

---

## [Editor Report · Decision Letter 1]

6 May 2025

A framework for identifying calcium accumulation problem in cropland: Integrating field surveys, legacy soil map, and machine learning models

PONE-D-25-00681R1

Dear Dr. An,

We’re pleased to inform you that your manuscript has been judged scientifically suitable for publication and will be formally accepted for publication once it meets all outstanding technical requirements.

Kind regards,

Vivek Sivakumar, Ph.D

Academic Editor

PLOS ONE

Additional Editor Comments (optional):

Dear Author,

The technical manuscript seems technically sound.
---

## [Editor Report · Acceptance letter]

PONE-D-25-00681R1

PLOS ONE

Dear Dr. An,

I'm pleased to inform you that your manuscript has been deemed suitable for publication in PLOS ONE. Congratulations! Your manuscript is now being handed over to our production team.

Kind regards,

on behalf of

Dr. Vivek Sivakumar

Academic Editor

PLOS ONE